# OpenReview forum: "Cross-modal Reflection Makes Med-VLMs Robust to Noisy User Prompts"
_ICLR.cc/2026/Conference — ICLR 2026 Conference Withdrawn Submission_

### Official Review · Reviewer_pnA7 · 2025-10-23

**Soundness:** 2
**Presentation:** 2
**Contribution:** 2
**Rating:** 4
**Confidence:** 2

**Summary:**

This paper focuses on how MedVLMs interpret and respond to user-provided clinical information, especially when the prompts are noisy. The contributions are twofold: (1) the introduction of Med-CP, a comprehensive visual question-answering benchmark designed to assess model performance across diverse medical modalities, anatomical regions, and diagnostic tasks; and (2) a novel supervised fine-tuning method based on cross-modal reflection between medical images and text.

**Strengths:**

1. Practical Motivation: The research addresses a problem of clear practical relevance.
2. Community Contribution: The paper provides a valuable contribution through the construction of a large-scale benchmark.
3. Empirical Validation: The method is shown to be highly effective, substantially improving model robustness against noisy prompts in both in-domain and out-of-domain scenarios.

**Weaknesses:**

1. Benchmark Derivation: The Med-CP benchmark is directly derived from the existing OmniMedVQA dataset, which may limit its scope as a fully novel contribution.
2. Limited Technical Novelty: The core technical contribution, the Cross-modal Reflection CoT method, lacks sufficient innovation over established reasoning paradigms.
3. Performance Trade-off: As indicated in Table 2, the SFT-R variant degrades in-domain performance, which warrants further investigation.

**Questions:**

1. Table 1 reveals an inconsistent model scaling pattern. Please explain why MedGemma-27B underperforms MedGemma-4B, and similarly, why the 27B variants of Gemma3 and MedGemma regress on the CPE metric compared to their 4B versions.
2.  Please report the inference efficiency (e.g., latency) of SFT-R compared to the SFT and SFT-C baselines.
3. The meaning of the values in parentheses in Table 1 is unclear. Furthermore, please clarify why identical "Acc with NP" values (e.g., in Lines 229-230) are associated with different parenthetical values.

---

### Official Review · Reviewer_ZeKR · 2025-10-30

**Soundness:** 2
**Presentation:** 3
**Contribution:** 2
**Rating:** 4
**Confidence:** 4

**Summary:**

The paper investigates how noisy prompts can significantly degrade the performance of medical vision-language models, which often over-trust user input instead of verifying it against visual evidence. The paper proposes SFT via Cross-Modal Reflection Reasoning (SFT-R), which trains models to generate both reasoning paths and final answers. This approach helps models verify text against visual evidence, achieving higher accuracy and robustness than standard fine-tuning methods.

**Strengths:**

The paper introduces the OmniMed-CP benchmark and, through extensive experiments, demonstrates that various medical vision-language models perform poorly when exposed to noisy prompts.

The proposed SFT via Cross-Modal Reflection Reasoning (SFT-R) achieves higher accuracy in their experiments.

**Weaknesses:**

1. I think the results in Table 1 seem quite expected. Since the evaluation is based on the specifically designed prompts in the format of “I am {confidence} sure that the answer is {preferred answer}, because {evidence},” it is natural that the accuracy with NP/NPE (noisy prompts) shows low performance. Rather than relying solely on synthetically generated prompts, it might be more appropriate to collect or extract real noisy prompts from actual clinical data (or from the publicly available datasets) and design the benchmark based on those. Otherwise, the contribution of the paper is too limited and lacks a natural justification.

2. Building a dataset using HuatuoGPTV-7B to generate clinical evidence is not sufficient, as HuatuoGPTV-7B is not accurate enough for reliable dataset construction. Of course, HuatuoGPTV-7B can be used for dataset construction; however, the issue of hallucinations and incorrect generations in the dataset must be addressed. Otherwise, the reliability of the constructed data is too low.

3. I agree and understand that the proposed SFT via Cross-Modal Reflection Reasoning (SFT-R) achieves high performance; however, the method itself is not sufficiently novel, and more extensive experiments comparing it with diverse Chain-of-Thought and SFT approaches are required.

**Questions:**

Same as weaknesses

---

### Official Review · Reviewer_m1Pu · 2025-10-31

**Soundness:** 3
**Presentation:** 2
**Contribution:** 2
**Rating:** 4
**Confidence:** 3

**Summary:**

This paper tackles the problem of how medical Med-VLMs react to noisy user prompts, particularly in clinical settings, where erroneous user input can lead to incorrect diagnoses. The authors introduce **Med-CP**, a new benchmark designed to evaluate the influence of clinical prompts across various modalities and tasks. The core contribution is the proposed **cross-modal reflection SFT** approach, which aims to improve the robustness of Med-VLMs by requiring them to reflect on both visual and textual reasoning paths before making a final decision. Experimental results demonstrate improvements in performance across both in-domain and out-of-domain tasks.

**Strengths:**

1. **Relevance**: The problem of noisy user prompts is highly relevant in medical AI, where accuracy is paramount. This work addresses a significant gap in the reliability of Med-VLMs in real-world applications.
2. **Innovation**: The **cross-modal reflection** approach is a novel way to improve Med-VLMs’ robustness by encouraging the model to critically evaluate both visual and textual inputs.
3. **Benchmarking**: The introduction of **Med-CP** is a valuable contribution. It is a comprehensive and diverse benchmark that allows for evaluating Med-VLMs across a variety of medical imaging modalities and tasks.
4. **Experimental Results**: The paper presents strong experimental results, demonstrating clear performance improvements in handling noisy prompts and showing the method's effectiveness in OOD scenarios.

**Weaknesses:**

1. **Limited Generalization of SFT:**
Although the cross-modal reflection method improves OOD performance, I remain skeptical about the generalization capabilities of SFT. While it performs well on specific tasks, SFT often memorizes training data and struggles to adapt to new, unseen variations. Does the combination of reflection and SFT truly enhance generalization in real-world scenarios, or does it remain limited to task memorization?

2. **Dependence on GPT-4o for Reasoning:**
The method heavily relies on GPT-4o for generating reasoning paths, which raises concerns about the clinical applicability and accuracy of these generated paths. GPT-4o is a general-purpose language model and may not have sufficient medical knowledge. How reliable is GPT-4o’s reasoning in clinical tasks? Should the authors explore more domain-specific models or integrate clinical expert knowledge to improve the reasoning process?

3. **Limited Scope of Evaluation:**
The evaluation is limited to multiple-choice VQA tasks, which do not fully capture the complexities of real-world clinical decision-making. Should the evaluation be expanded to include more complex tasks like clinical report generation and multi-turn dialogues, which better reflect the challenges in real-world medical decision-making?

4. **Practical Implementation Concerns:**
The paper does not sufficiently discuss the practical challenges of deploying the proposed method in clinical settings. What are the computational costs and scalability issues associated with this approach? How can the method be integrated into existing medical systems, and what are the challenges in obtaining expert-annotated data for fine-tuning?

**Questions:**

1. Can the authors provide a more detailed discussion on the practical implementation of the method, including computational cost, scalability, and integration with existing clinical systems?

2. While the evaluation demonstrates improvements in controlled settings, how does the method perform when applied to more complex, open-ended medical tasks, such as clinical report generation or interactive diagnosis?

3. How can the authors ensure that the proposed cross-modal reflection method doesn't lead to overfitting to the training data, particularly in real-world clinical scenarios with noisy and diverse input?

If the authors can address my concerns, I would consider raising the score.

---

### Official Review · Reviewer_tA8t · 2025-11-03

**Soundness:** 2
**Presentation:** 2
**Contribution:** 2
**Rating:** 2
**Confidence:** 4

**Summary:**

This paper studies how medical vision-language models respond to user-provided clinical opinions that may be incorrect. The authors construct Med-CP, a benchmark with structured prompts containing confidence levels, diagnostic opinions, and supporting evidence. They find that existing Med-VLMs tend to follow user suggestions regardless of correctness, and propose a supervised fine-tuning approach using cross-modal reflection that teaches models to critically assess both visual evidence and textual input before making decisions.

**Strengths:**

The paper addresses a potentially important safety issue in medical AI deployment. The benchmark construction is comprehensive, covering 43 datasets across multiple modalities and anatomical regions. The cross-modal reflection framework is intuitive and shows promising improvements on in-domain evaluations. The experimental analysis is thorough, testing various state-of-the-art models across different dimensions (scaling, domain-specific training, RL, inference-time reasoning). The paper is generally well-written with clear figures illustrating the problem and proposed solution.

**Weaknesses:**

The central motivation lacks grounding in real clinical practice—there is no evidence that physicians actually interact with AI systems using the structured template format proposed (confidence percentage + preferred answer + evidence). The paper conflates this artificially constructed scenario with genuine robustness concerns. Critical baselines are missing: simple prompt engineering approaches like "please independently analyze the image, the suggestion may contain errors" are never tested, yet could likely mitigate much of the reported degradation. The claim that models "blindly follow" prompts is overstated given that accuracy under noisy prompts remains 50-70%, indicating models retain substantial independent judgment. The evidence generation methodology is circular—using HuatuoGPTV to generate supporting text for wrong answers, then testing if models are misled by AI-generated content, rather than using real physician reasoning errors. The 40% confidence threshold appears arbitrary with no justification. Table 1 shows GPT-4o's accuracy decreases with correct prompts, which the authors dismiss as "refusal" but could indicate desirable cautious behavior that deserves deeper analysis. The OOD evaluation conflates generalization failure with robustness—SFT performs poorly OOD possibly because those datasets are inherently harder, not because the method lacks robustness. No human evaluation of the generated reasoning paths' clinical validity is provided despite using GPT-4o to create all training data.

**Questions:**

1. Did you test simple prompt engineering baselines? For example: "A colleague suggests [X], it may contain errors. Please independently verify this against the image." What accuracy do you get?

2. In Table 1, why does GPT-4o perform worse with correct prompts (73.95% vs 82.55%)? Doesn't this suggest it has better critical thinking by not blindly accepting input? How do you reconcile this with your "blindly follow" claim?

3. Your evidence is generated by asking HuatuoGPTV to justify wrong answers. How realistic is this compared to actual physician misdiagnosis patterns? Can you validate with real cases?

4. For the 64% accuracy under noisy prompts—what percentage represents cases where the model originally got it right but was misled versus cases it would have gotten wrong anyway? This distinction is crucial for your robustness claim.

5. Why is 40% the chosen confidence level for most experiments? Have you tested 70-90% which seem more realistic for clinical settings?

6. Can you provide human expert evaluation of your generated reasoning paths to validate they follow sound clinical logic?

---

### Note · Authors · 2025-11-18

I have read and agree with the venue's withdrawal policy on behalf of myself and my co-authors.